# Interfacial Microstructure Produced during Dissimilar AA6013/Ti-6Al-4V Friction Stir Lap Welding under Zero-Penetration Condition

**Alexander Kalinenko [1], Pavel Dolzhenko [1], Sergey Malopheyev [1], Diana Yuzbekova [1], Yuliya Borisova [1], Ivan Shishov [2], Vasiliy Mishin [2], Sergey Mironov [1,\* ] and Rustam Kaibyshev [1]**

[1] Laboratory of Mechanical Properties of Nanoscale Materials and Superalloys, Belgorod National Research University, 308015 Belgorod, Russia; kalinenko@bsu.edu.ru (A.K.); dolzhenko_p@bsu.edu.ru (P.D.); malofeev@bsu.edu.ru (S.M.); yuzbekova@bsu.edu.ru (D.Y.); borisova_yu@bsu.edu.ru (Y.B.); rustam_kaibyshev@bsu.edu.ru (R.K.)

[2] Institute of Machinery, Materials, and Transport, Peter the Great St. Petersburg Polytechnic University, 195251 St. Petersburg, Russia; shishov_ia@spbstu.ru (I.S.); mishin_vv@spbstu.ru (V.M.)

\* Correspondence: mironov@bsu.edu.ru; Tel.: +7-4722-585456

**Abstract:** The purpose of this study was to investigate the interfacial microstructure that was produced during dissimilar friction stir lap welding (FSW) of 6013 aluminum alloy and Ti-6Al-4V. FSW was conducted under a zero-penetration condition, i.e., the welding tool was plunged exclusively into the upper (aluminum) plate of the dissimilar lap joint. To facilitate the interpretation of microstructural processes, finite element modeling (FEM) was applied to evaluate the temperature field within the weld zone. The FEM simulation revealed a very sharp temperature gradient across the dissimilar interface. This effect was attributed to the generation of FSW heat exclusively within the aluminum part and a relatively low thermal conductivity of titanium. The abrupt temperature drop on the titanium side imposed a strict limitation on the diffusion penetration of aluminum and thus resulted in a relatively thin (~0.5 μm) and discontinuous intermetallic compound. Due to the complex chemical composition of the FSWed aluminum alloy, the diffusion processes also involved alloying elements. Consequently, the evolved intermetallic compound had a complicated chemical composition, with the principal elements being aluminum, titanium, silicon, manganese, and magnesium.

**Keywords:** dissimilar Al/Ti friction stir welding; scanning electron microscopy (SEM); finite element modeling (FEM); microstructure

## 1. Introduction

Due to their attractive combination of service properties, aluminum and titanium alloys are widely used in the transportation industry. In some engineering applications, the mutual joining of these two materials is desirable. Unfortunately, the application of conventional fusion techniques for this purpose is challenging owing to the distinct incompatibility of the thermal properties of these metals. Hence, the innovative *solid-state* friction stir welding (FSW) technology is often considered a promising candidate.

Extensive research over the last two decades has demonstrated the feasibility of FSW for dissimilar joining of aluminum and titanium alloys. However, it has been found that one of the most significant problems in this area is the formation of intermetallic compounds at the welded interface [1–3]. Being naturally brittle, the intermetallics promote cracking, thus leading to the premature failure of dissimilar FSW joints [4–14].

Typically, the formation of TiAl₃ intermetallic is reported during the dissimilar FSW of aluminum and titanium alloys [4–26]. This observation is usually explained in terms of

the lowest free energy of this compound at the typical FSW conditions. In some cases, however, the intermetallic layer may have a composite structure including TiAl$_3$ and TiAl compounds [8,15,19,24,27].

To inhibit the intermetallic reaction between aluminum and titanium during FSW, interlayer materials are sometimes used [13,16–18,22,28,29]. These include zinc [18,28,29], copper [13,17], niobium [16], or even carbon fiber-reinforced polymer [22]. However, due to the extensive fragmentation of the interlayer during FSW, this approach does not seem to be entirely effective.

It has been well established that the thickness of the intermetallic layer is closely linked to the FSW heat input [5,8,9,19,21,26,30]. Hence, a number of strategies have been elaborated to optimize this characteristic in order to enhance the weld strength. The relatively simple approaches include a control of the tool rotation rate [5,19,27,31–33] or diameter of the tool shoulder [33], or involve the application of the stationary shoulder FSW [34]. A more specific approach represents the control of the tool offset/position during FSW [5,6,11,12,17,18,26,30,35–37]. It has been found that shifting the welding tool towards the aluminum side is very efficient for reducing heat generation and thus shrinking the intermetallic layer [11,12,30].

In the previous study, an extreme case of the latter approach was applied [38]. Specifically, the welding tool was inserted solely into the aluminum part of the dissimilar joint in a lap welding configuration. It was found that the produced joints exhibited excellent performance during transverse tensile tests. Particularly, those were failed in the aluminum part, thus implying that the interfacial structure was not a critical issue.

The present study focused on the examination of the interfacial microstructure that evolved during FSW under such a zero-penetration condition. To facilitate microstructural analysis, finite element modeling (FEM) was applied to simulate the thermal conditions within the stir zone (A brief background on the FEM simulation of the FSW thermal cycle is provided in Section 3.1).

## 2. Materials and Methods

The program materials employed in the present study included commercial 6013 aluminum alloy and Ti-6Al-4V titanium alloy. These two materials are widely used in the transportation industry, and thus their practical usage should perhaps benefit from the application of FSW for their mutual joining. Moreover, the FSW behavior of these two alloys has been studied relatively well.

The aluminum alloy was produced by semi-continuous casting using laboratory casting equipment. The cast ingot was homogenized at 550 °C for 4 h and then cold rolled to an 80% thickness reduction. The Ti-6Al-4V alloy was supplied in a mill-annealed condition.

The 2-mm-thick aluminum and titanium sheets were friction stir lap welded using an AccuStir FSW machine. To provide a suitable quality for the surfaces to be welded, they were ground using sandpaper of 500 emery grit and then degreased with acetone. In all cases, the aluminum workpiece was placed on the top side of the dissimilar joint, while the titanium one was placed on its bottom side. To maintain consistency with the previous work [38], the welding tool was plunged only into the aluminum part. Furthermore, FSW was conducted in a plunge depth control mode, and the distance between the probe tip and the titanium part was kept at ≈50 μm. The welding tool was manufactured from tool steel and consisted of a concave-shaped shoulder 12.5 mm in diameter and an M5 threaded probe 1.9 mm in length. In order to provide a suitable welding condition [38], FSW was conducted at a tool rotation rate of 1100 rpm and a tool travel speed of 3 inches per minute (≈76.2 mm/min). All welding trials were performed at a tool tilt angle of 2.5° and using a stainless steel backing plate. The magnitude of the Z-force during the steady welding stage was measured at ~4.5 kN. A typical convention for FSW geometry was adopted, with WD, ND, and TD being the welding direction, normal direction, and transverse direction, respectively.

To record the weld thermal cycle, a series of K-type thermocouples were placed at the aluminum–titanium interface. To evaluate the temperature distribution, the thermocouples were located in a range from 0.1 to 5.35 mm from the edge of the welding tool on both the retreating and advancing sides of the weld zone.

Microstructural examinations were conducted using scanning electron microscopy (SEM), energy-dispersive X-ray spectroscopy (EDS), and electron backscatter diffraction (EBSD). To obtain a suitable surface finish, microstructural samples were cut perpendicular to the welding direction of the welded joints and hot-mounted into an electro-conductive resin. Then, they were mechanically polished in a conventional fashion using a series of water-abrasive papers and diamond pastes. The final polishing step comprised 24-h vibratory polishing using commercial OPS suspension.

All microstructural observations were performed with an FEI Quanta 600 field emission gun SEM. The microscope was equipped with the TSL OIM™ system and operated at an accelerated voltage of 20 kV. EBSD maps were collected using a hexagonal scanning grid and a scan step size of 0.1 μm. For each diffraction pattern, nine Kikuchi bands were used for indexing to minimize errors. To avoid any modification of the experimental EBSD data, no post-processing (or clean-up) procedure was applied.

## 3. FEM Simulation

Considering the complexity of the underlying microstructural processes, the numerical simulation of the formation of intermetallic compounds during FSW is challenging. Hence, the FEM approach in the present study was applied only for a simulation of the *thermal field* generated within the weld zone during FSW. The obtained results are summarized in the present section.

### 3.1. Broad Aspects of the Simulation Model

The unique characteristic of FSW is extremely large true strains, which may reach a magnitude of ~80 [39]. Accordingly, the application of the conventional Lagrangian FEM approach for simulation of the thermo-mechanical aspects of FSW is challenging because of the severe distortion of meshing elements [40–42]. To achieve acceptable accuracy of calculations in this case, it is necessary to use meshes with an extremely small element size, and to perform frequent re-meshing during the analysis. Both of these factors lead to very high computational costs and, consequently, the simulation process becomes excessively time-consuming. Thus, an alternative Eulerian FEM approach, which implies a free material flow through the mesh without element distortion, is often considered as a more suitable technique for this purpose [43–45]. In practice, both Lagrangian and Eulerian techniques are combined in order to yield the best result [46–49].

In the present study, the coupled Eulerian–Lagrangian (CEL) approach was utilized. It allowed for considering an interaction between the Eulerian and Lagrangian bodies within the same FEM model. The thermo-mechanical 3D model of dissimilar friction stir lap welding of AA6013 and Ti-6Al-4V was elaborated using the commercial Abaqus 2020 software.

The aluminum workpiece was modeled as an Eulerian body, while the titanium workpiece and FSW tool were considered as Lagrangian bodies (Figure 1).

The size of the Ti-6Al-4V plate and AA6013 workpiece was taken to be $70 \times 70 \times 2$ mm$^3$ (length × width × thickness), while the total size of the Eulerian part was $70 \times 70 \times 4$ mm$^3$. The difference between the workpiece and the Eulerian domain was left empty in order to visualize the material flow during FSW. The backing plate was neglected in the model.

The aluminum workpiece was meshed with 59,640 EC3D8RT (eight-node thermally coupled linear Eulerian brick) elements. To save computational time, a different mesh size was used within the stir zone and the remaining part of the aluminum workpiece. Specifically, the mesh size within the stir zone was $1.0 \times 0.5 \times 0.5$ mm$^3$ (length × width × thickness), while that in the remaining aluminum workpiece was $1.0 \times 1.0 \times 1.0$ mm$^3$

(Figure 1). The Ti-6Al-4V workpiece was meshed using C3D8T (eight-node thermally coupled brick) elements with a mesh size of 1.0 × 1.0 × 1.0 mm³.

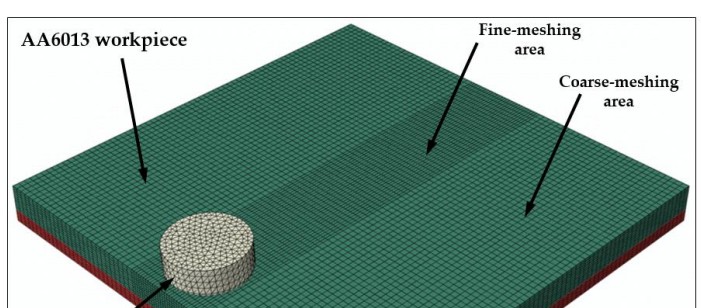

**Figure 1.** The meshing model of the FEM simulation of the FSW process. WD, TD, and ND are welding, transverse, and normal directions, respectively.

It is known that the design of the FSW tool may exert an essential influence on computational accuracy in FEM simulations; moreover, the simplification of the tool geometry may lead to a significant underestimation of the accumulated strain [50]. To avoid this problem, the design of the simulated tool in the present study completely replicated the real one (including probe threads). The tool was meshed using 7956 C3D10MT (10-node thermally coupled tetrahedron) elements with an approximate size of 0.87 mm.

The titanium plate was fully constrained against motion in all directions. Moreover, to prevent the motion of the material of the aluminum workpiece outside the computational domain, the boundary conditions on its bottom and side surfaces were set to zero velocities.

The elaborated model included two stages of the FSW process. During the first one, the welding tool was rotated with an angular velocity of 1100 rpm and then inserted into the aluminum workpiece. The distance between the probe tip of the plunged tool and the titanium workpiece was kept at 50 μm. During the second stage, the rotating tool was translated along the aluminum workpiece with a feed rate of 76.2 mm/min. In all cases, the tool tilting angle of 2.5° was also taken into consideration in the model.

### 3.2. Friction Condition at the Tool/Workpiece Interface

The contact condition between the welding tool and the workpiece was simulated using the classical Coulomb friction law:

$$\tau = \mu p \tag{1}$$

where $\tau$ is the frictional shear stress, $\mu$ is the friction coefficient, and $p$ is the contact pressure between the tool and workpiece.

Considering the complexity of the physical processes involved in FSW, the determination of the friction coefficient is challenging. In the practice of the simulation of FSW of aluminum alloys, the friction coefficient is often assumed to be constant, and typically lies in the range of 0.25 to 0.5 [51–54]. In the present study, the friction coefficient was taken to be 0.35.

### *3.3. Thermal Conditions*

In the model, it was assumed that the heat generation arose from (i) the friction between the rotating tool and the welded material, and (ii) the plastic deformation of the aluminum workpiece [55]. To simplify thermal analysis, heat dissipation was assumed to be governed by heat transfer, while the effects of thermal convection and thermal radiation were neglected.

Given the sensitivity of the heat transfer process to the local variation of contact conditions (surface roughness, pressure, etc.), an elucidation of the heat transfer coefficient is usually challenging. To avoid this difficulty, the coefficients were selected in order to provide the best fit between the experiment and the model. Specifically, it was found that the peak welding temperature was mainly determined by the heat transfer coefficient between aluminum and titanium. Meanwhile, the cooling rate was mainly influenced by the heat transfer coefficient between titanium and the backing plate. In this way, the heat transfer coefficient between the aluminum and titanium workpieces was taken to be 2000 W (m² × °C), while that between the titanium and the backing plate was assumed to be 1000 W (m² × °C).

### *3.4. Material Model*

The thermo-mechanical behavior of the aluminum workpiece was simulated using the classical Johnson–Cook equation [56]:

$$\sigma = \left(A + B\varepsilon^n\right) \times \left[1 + C\ln\left(1 + \frac{\dot{\varepsilon}}{\dot{\varepsilon}_0}\right)\right] \times \left[1 - \left(\frac{T - T_r}{T_m - T_r}\right)^m\right] \tag{2}$$

where $\sigma$ is flow stress, $\varepsilon$ is strain, $\dot{\varepsilon}$ is strain rate, $\dot{\varepsilon}_0$ is the normalized strain rate, $T$ is temperature, $T_r$ is the room temperature (taken to be 25 °C), $T_m$ is the incipient melting temperature, and $A$, $B$, $n$, and $m$ are the material constants.

The input parameters for the Johnson–Cook equation are summarized in Table 1. Except for the temperature exponent, the parameters were taken from a recent study by Saloomi [47]. The preliminary analysis showed that a value of $m$ = 1.34 (i.e., similar to that in Ref. [47]) leads to a significant overestimation of the Z-force. Meanwhile, a relatively good fit between the calculations and measurements was obtained at $m$ = 0.4 (Table 1). It is suggested that this effect was associated with a replication of the real design of the welding tool (with a threaded tool probe) in the present study.

The physical properties of the model materials were taken from Refs. [57–60] and shown in Table 2.

**Table 1.** Material constants of 6013 aluminum alloy for the Johnson–Cook equation.

| Constant | Definition | Magnitude | Unit |
|:---:|:---:|:---:|:---:|
| A | Yield stress at ambient temperature | 324 | MPa |
| B | Strain factor | 114 | MPa |
| n | Strain exponent | 0.42 | - |
| C | Strain rate factor | 0.002 | - |
| $\dot{\varepsilon}_0$ | Normalized strain rate | 1 | - |
| $T_m$ | Incipient melting temperature | 582 | °C |
| m | Temperature exponent | 0.4 | - |

**Table 2.** Physical properties of the model materials.

| Material | Density, kg/m³ | Thermal Conductivity, W/(m × K) | Specific Heat, J/(kg × K) | Young's Modulus, GPa | Poisson's Ratio |
|---|---|---|---|---|---|
| AA6013 | 2750 | 161 | 945 | 60 | 0.3 |
| Ti-6Al-4V | 4520 | 20.4 | 523 | 122 | 0.32 |

*3.5. Model Validation*

To verify the FEM model, the computed thermal cycles were compared with experimental observations (Figure 2). The relatively good fit between the results suggested the reliability of the elaborated model.

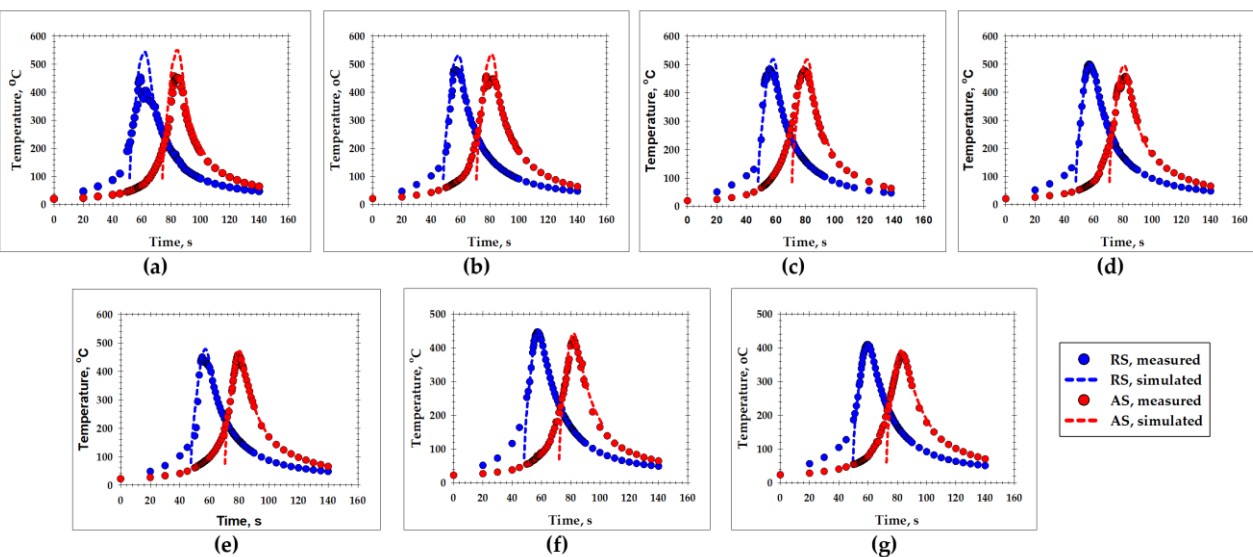

**Figure 2.** Validation of the FEM model: the comparison of the measured and simulated thermal cycles as a function of a distance from the surface of the tool probe: (**a**) 0.1 mm, (**b**) 0.85 mm, (**c**) 1.6 mm, (**d**) 2.35 mm, (**e**) 3.1 mm, (**f**) 3.85 mm, and (**g**) 5.35 mm. RS and AS denote the retreating and advancing sides, respectively.

*3.6. The Simulated Temperature Field within the Weld Zone*

The simulated temperature distribution within the entire weld is shown in Figure 3a. To make a close inspection of the temperature field within the stir zone, the transverse cross-section of the welded joint directly behind the welding tool is shown in Figure 3b.

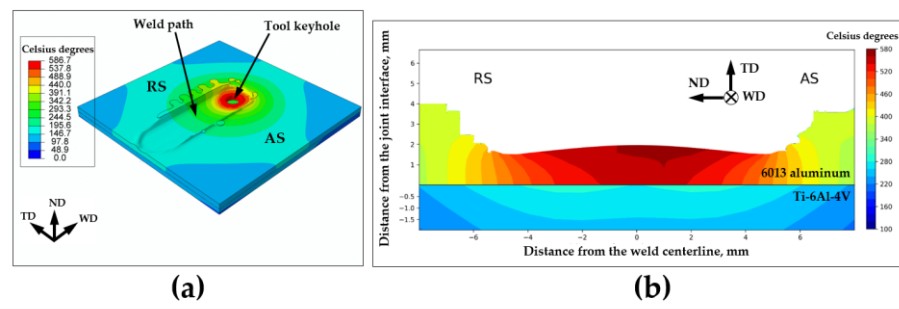

**Figure 3.** The simulated temperature distribution: (**a**) within the entire weld, and (**b**) in the transverse cross section of the weld zone, directly behind the welding tool. WD, TD, and ND are welding, transverse, and normal directions, respectively. RS and AS denote the retreating side and advancing side, respectively.

A very sharp temperature gradient across the aluminum/titanium interface can be seen. Specifically, the temperature on the aluminum side of the interface exceeded 500 °C while that on the titanium side was well below 350 °C (Figure 3b). This effect is due to the relatively low thermal conductivity of Ti-6Al-4V (Table 2).

## 4. Experimental Results

### 4.1. SEM Observations

A typical SEM micrograph taken from the welded surface between 6013 aluminum alloy and Ti-6Al-4V titanium alloy is shown in Figure 4a. The development of a relatively thin (~0.5 μm), gray-colored layer at the joint interface is seen (highlighted by arrows). The magnified image of the interfacial layer is given in Figure 4b.

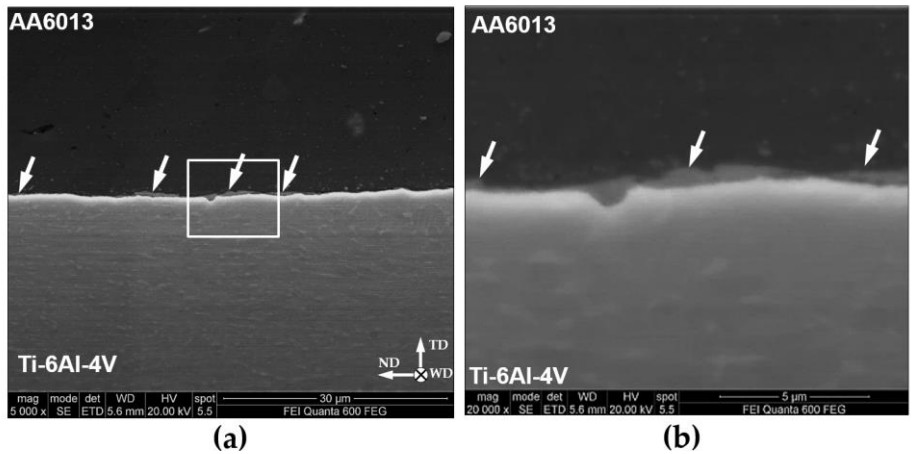

**Figure 4.** (**a**) Typical SEM images of the welded surface between 6013 aluminum alloy and Ti-6Al-4V with the selected area given at higher magnification in (**b**). Arrows indicate the interphase layer. ND, WD, and TD are normal direction, welding direction, and transverse direction, respectively.

In terms of the SEM contrast, this layer was broadly similar to the $TiAl_3$ compound, which develops in Al/Ti sandwich structures during long-term annealing at 500 °C (e.g., [61]). In the present study, a notable characteristic of the interfacial layer was its complex topography (Figure 4b). Of particular importance was the observation that the interfacial layer was *discontinuous* in nature (Figure 4a).

### 4.2. EDS Analysis

To investigate the elemental composition of the interfacial layer, EDS mapping was applied. The typical results are shown in Figure 5.

It was found that this layer was enriched by silicon, manganese, and perhaps magnesium. This interesting observation allows for the deduction of the following two conclusions. First, as the welded materials are not pure metals but alloys, FSW may promote not only the mutual diffusion of aluminum and titanium but also the diffusion of the alloying elements. Second, if the same alloying elements are present in both welded alloys, such diffusion may result in their accumulation at the joint interface.

To allow a closer inspection of the interfacial layer, quantitative EDS analysis was utilized. The typical results are shown in Figure 6.

It was found that the chemical composition of the FSW-induced interfacial layer was close to $TiAl_3$ intermetallic. This observation was in agreement with numerous reports in the FSW literature [4–26]. However, it is important to emphasize that the evolved intermetallic was heavily alloyed with silicon, manganese, and magnesium. Moreover, an increased concentration of iron and chromium was also found. Hence, it can be suggested

that the properties of the FSW-induced compound may essentially differ from the properties of TiAl₃ intermetallic.

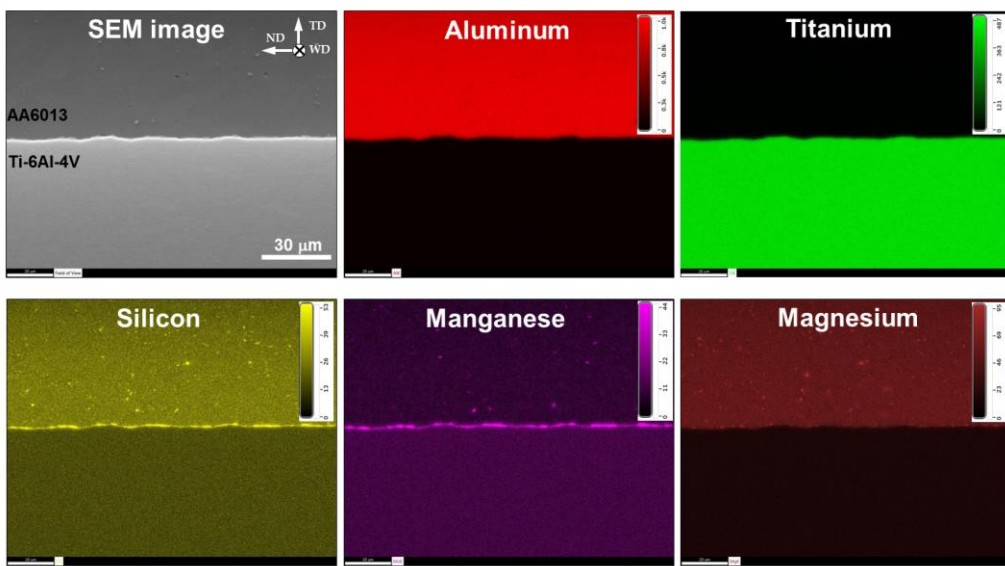

**Figure 5.** SEM images and EDS maps showing elemental distribution across the welded surface. ND, WD, and TD are normal direction, welding direction, and transverse direction, respectively.

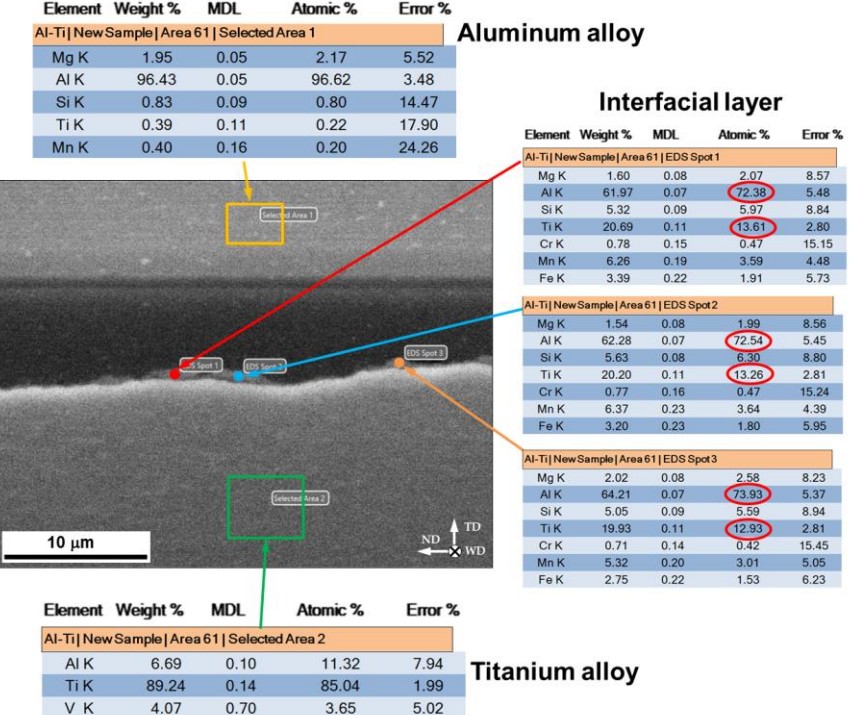

**Figure 6.** Quantitative EDS analysis of welded materials and interfacial layer. ND, WD, and TD are normal direction, welding direction, and transverse direction, respectively.

### 4.3. EBSD Measurements

In an attempt to gain additional insights into the intermetallic layer, EBSD maps were taken from the welded surface. A typical example is shown in Figure 7a. The magnified image of the interfacial microstructure is given in Figure 7b.

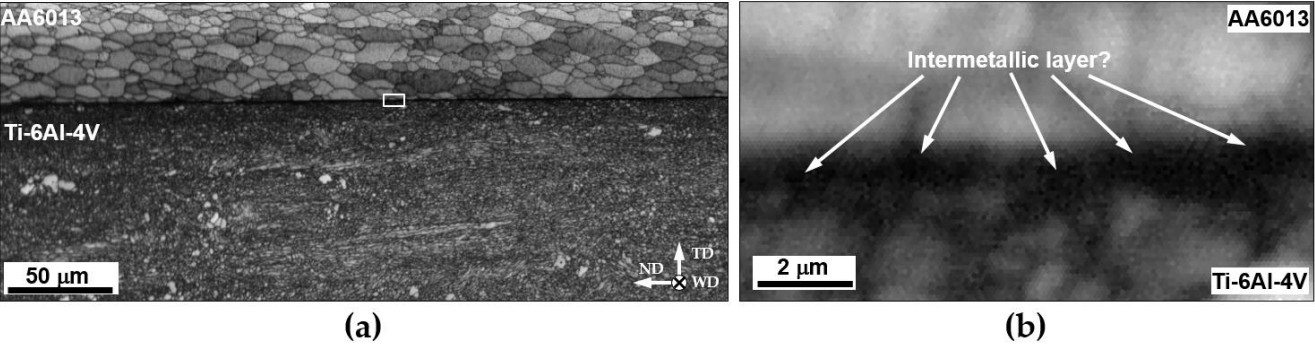

**Figure 7.** EBSD Kikuchi band contrast map taken from the welded surface: (**a**) low-magnification overview with selected area shown at higher magnification in (**b**). ND, WD, and TD are normal direction, welding direction, and transverse direction, respectively.

It was found that the microstructure on the aluminum side of the dissimilar joint was characterized by relatively coarse (~10 μm) recrystallized grains (Figure 7a). The development of this microstructure was presumably a result of a comparatively high temperature within the aluminum stir zone (Figure 3b) as well as a low cooling rate (Figure 2a). Meanwhile, the microstructure on the titanium side of the joint was much finer-grained and heavily deformed in appearance (Figure 7a). This observation was in accordance with the FEM simulation, which predicted a relatively low temperature on the titanium side (Figure 3b). Unfortunately, no indexable Kikuchi patterns were detected *within* the intermetallic layer. Accordingly, this layer appeared black in the EBSD map, and no information on its internal structure was obtained (Figure 7b). It was only found that the intermetallic particles were comparable in size to the adjoining titanium grains (Figure 7b). Hence, it was suggested that the development of the intermetallic layer may be influenced by the titanium grain structure.

## 5. Discussion

One of the most significant results of the FEM simulation was a sharp temperature gradient across the Al/Ti interface (Figure 3b). Specifically, the temperature on the aluminum side of the dissimilar joint was as high as ~0.9 $T_m$ (where $T_m$ is the incipient melting point), while that on the titanium side was only ~0.3 $T_m$ (Figure 3b). This phenomenon was obviously attributable to the following two factors: (i) the generation of FSW heat exclusively within the aluminum part, and (ii) the comparatively low thermal conductivity of titanium alloy (Table 2), which resulted in a slow heat transfer through the Al/Ti interface. In turn, both of these factors were a direct result of the zero-penetration strategy applied in the present study.

Given the drastic temperature difference between the aluminum and titanium parts, it is highly likely that the formation of the intermetallic compound was governed by a diffusion of aluminum into titanium but not vice versa. If so, the abrupt temperature drop within the titanium plate (Figure 3b) should naturally restrict the thickening of the intermetallic layer. Thus, the formation of the comparatively thin intermetallic layer in the present study (Figure 4) was perhaps also a direct consequence of the zero-penetration approach that was employed.

Another important issue was the complex topography of the intermetallic layer and, particularly, its discontinuous nature (Figure 4). From a broad perspective, such observations are usually attributable to local variations in temperature and strain. However, considering the fine-scale character of the intermetallic particles, it was likely that their local growth was influenced by diffusion anisotropy due to the particular crystallographic orientation of titanium grains (Figure 7b).

## 6. Summary

This study was undertaken to examine the interfacial microstructure that was produced during dissimilar friction stir lap welding of aluminum and titanium alloys using the zero-penetration approach. This approach was realized by plunging the FSW tool exclusively into the upper (aluminum) plate of the dissimilar lap joint, while the distance between the tip of the tool probe and the titanium plate was kept at ~50 μm. To facilitate the interpretation of microstructural observations, the FEM technique was applied to simulate the thermal field evolved during FSW within the weld zone. The main conclusions derived from the present study were as follows:

(1) The FEM simulation revealed a very sharp temperature gradient across the Al/Ti interface. Specifically, the welding temperature on the aluminum side of the dissimilar joint was as high as 0.9 $T_m$, while that on the titanium side was as low as 0.3 $T_m$. This result was attributable to the (i) generation of the weld heat exclusively within the aluminum part, and (ii) the relatively low thermal conductivity of titanium, which provided a slow heat transfer from aluminum to titanium. As a result of the distinct temperature difference between the dissimilar parts, the development of the interfacial microstructure was governed by the diffusion of aluminum into titanium, but not vice versa. Hence, the evolved intermetallic compound was close to $TiAl_3$.

(2) Due to the complex chemical composition of the FSWed aluminum alloy, the diffusion processes also involved the alloying elements, mainly silicon, manganese, and magnesium. Accordingly, the evolved intermetallic compound had a complicated chemical composition, with the principal elements being aluminum, titanium, silicon, manganese, and magnesium.

(3) The intermetallic compound was only ~0.5 μm in thickness and had a discontinuous character. These observations were attributed to the abrupt temperature drop within the titanium plate, which restricted the diffusion distance of aluminum.

**Author Contributions:** Conceptualization, S.M. (Sergey Mironov); methodology, A.K., P.D., D.Y., Y.B., S.M. (Sergey Malopheyev), I.S. and V.M.; software, I.S. and V.M.; validation, A.K. and S.M. (Sergey Mironov); formal analysis, S.M. (Sergey Mironov); investigation, A.K., P.D., D.Y., Y.B., S.M. (Sergey Malopheyev), I.S., V.M. and S.M. (Sergey Mironov); resources, R.K.; data curation, S.M. (Sergey Mironov); writing—original draft preparation, S.M. (Sergey Mironov); writing—review and editing, A.K., P.D., D.Y., Y.B., S.M. (Sergey Malopheyev), I.S., V.M. and R.K.; visualization, S.M. (Sergey Mironov); supervision, R.K.; project administration, S.M. (Sergey Mironov). All authors have read and agreed to the published version of the manuscript.

**Funding:** This research was funded by the Russian Science Foundation, grant No. 22-49-04401.

**Data Availability Statement:** Data will be made available on request.

**Acknowledgments:** The experimental work was performed using the equipment of the Research Equipment Sharing Center "Technology and Materials" at Belgorod National Research University (financial support from the Ministry of Science and Higher Education of the Russian Federation under agreement No. 075-15-2021-690, unique project identifier RF 2296.61321X0030).

**Conflicts of Interest:** The authors declare no conflicts of interest.

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
