# Peer review of "Interfacial Microstructure Produced during Dissimilar AA6013/Ti-6Al-4V Friction Stir Lap Welding under Zero-Penetration Condition"

_metals, doi:10.3390/met13101667_

Round 1
Reviewer 1 Report
This is a review of a manuscript submitted to Metals entitled: "Interfacial microstructure produced during dissimilar AA6013/Ti-6Al-4V friction-stir lap welding under zero-penetration condition". The paper presents a study on dissimilar friction stir welding (FSW) of AA6013/Ti-6Al-4V in overlap joint configuration with regards to FEM simulation of temperature and formatoin of intermetallic compound at the interface during FSW, and it should enrich the knowledge in this field.
The paper is well-written and interesting to read. However, reviewer would suggest authors to add literature studies/background on simulation of the thermal cycle during FSW in introduction part.
Author Response
Responses to the comments of Reviewer #1
on the paper entitled “Interfacial microstructure produced during dissimilar AA6013/Ti-6Al-4V friction-stir lap welding under zero-penetration condition” (Manuscript ID: metals-2614077)
The authors would like to express their gratitude to Reviewer for his/her remarks. Below, we provided specific replies to the issues raised.
Note: Reviewer’s comments are highlighted with bold.
This is a review of a manuscript submitted to Metals entitled: "Interfacial microstructure produced during dissimilar AA6013/Ti-6Al-4V friction-stir lap welding under zero-penetration condition". The paper presents a study on dissimilar friction stir welding (FSW) of AA6013/Ti-6Al-4V in overlap joint configuration with regards to FEM simulation of temperature and formatoin of intermetallic compound at the interface during FSW, and it should enrich the knowledge in this field.
The paper is well-written and interesting to read. However, reviewer would suggest authors to add literature studies/background on simulation of the thermal cycle during FSW in introduction part.
Authors’ response
According to the comment, a brief background on the numerical simulation of the FSW thermal cycle has been provided in the revised manuscript. Given the specific character of this analysis (which is distinctly different from the current version of the introduction section), it was provided in a FEM section of the revised manuscript (Page 3, Section 3.1, Paragraph 1):
"The unique characteristic of FSW is the extremely large true strains, which may reach a magnitude of ~80 [39]. Accordingly, the application of the conventional Lagrangian FEM approach for simulation of thermo-mechanical aspects of FSW is challenging because of the severe distortion of meshing elements [40-42]. To achieve acceptable accuracy of calculations in this case, it is necessary to use meshes with extremely small element size and to perform frequent re-meshing during the analysis. Both of these factors lead to very high computational costs and, consequently, the simulation process becomes excessively time-consuming. Thus, an alternative Eulerian FEM approach, which implies a free material flow through the mesh without element distortion, is often considered a more suitable technique for this purpose [43-45]. In practice, both Lagrangian and Eulerian techniques are combined together in order to yield the best result [46-49]. "
References:
"[39] Kuykendall, K.; Nelson, T.; Sorensen, C. On the Selection of Constitutive Laws Used in Modeling Friction Stir Welding. Int. J. Mach. Tools Manuf. 2013, 74, 74–85. https://doi.org/10.1016/j.ijmachtools.2013.07.004.
[40] Arora, A.; Zhang, Z.; De, A.; DebRoy, T. Strains and Strain Rates during Friction Stir Welding. Scripta Mater. 2009, 61, 863–866. https://doi.org/10.1016/j.scriptamat.2009.07.015.
[41] Li, Z.; Ding, H.; Chen, Y.; Li, J.; Liu, L. Strain Accumulation and Microstructural Evolution During Friction Stir Welding of Pure Magnesium. Front. Mater. 2020, 7. https://doi.org/10.3389/fmats.2020.603464.
[42] Ma, X.; Xu, S.; Wang, F.; Zhao, Y.; Meng, X.; Xie, Y.; Wan, L.; Huang, Y. Effect of Temperature and Material Flow Gradients on Mechanical Performances of Friction Stir Welded AA6082-T6 Joints. Materials. 2022, 15, 6579. https://doi.org/10.3390/ma15196579.
[43] Svensson, D.; Andersson, T.; Lassila, A.A. Coupled Eulerian–Lagrangian Simulation and Experimental Investigation of Indexable Drilling. Int. J. Adv. Manuf. Technol. 2022, 121, 471–486. https://doi.org/10.1007/s00170-022-09275-0.
[44] Ducobu, F.; Riviere-Lorphevre, E.; Filippi, E. Application of the Coupled Eulerian-Lagrangian (CEL) Method to the Modeling of Orthogonal Cutting. Eur. J. Mech. - A/Solids. 2016, 59, 58–66. https://doi.org/10.1016/j.euromechsol.2016.03.008.
[45] Ko, J.; Jeong, S.; Kim, J. Application of a Coupled Eulerian-Lagrangian Technique on Constructability Problems of Site on Very Soft Soil. Appl. Sci. 2017, 7, 1080. https://doi.org/10.3390/app7101080.
[46] Guerdoux, S.; Fourment, L. A 3D Numerical Simulation of Different Phases of Friction Stir Welding. Model. Simul. Mater. Sci. Eng. 2009, 17, 075001. https://doi.org/10.1088/0965-0393/17/7/075001.
[47] Salloomi, K.N. Fully Coupled Thermomechanical Simulation of Friction Stir Welding of Aluminum 6061-T6 Alloy T-joint. J. Manuf. Process. 2019, 45, 746–754. https://doi.org/10.1016/j.jmapro.2019.06.030.
[48] Al-Badour, F.; Merah, N.; Shuaib, A.; Bazoune, A. Coupled Eulerian Lagrangian Finite Element Modeling of Friction Stir Welding Processes. J. Mater. Process. Technol. 2013, 213, 1433–1439. https://doi.org/10.1016/j.jmatprotec.2013.02.014.
[49] Meyghani, B.; Awang, M.B.; Wu, C.S. Thermal Analysis of Friction Stir Processing (FSP) Using Arbitrary Lagrangian‐Eulerian (ALE) and Smoothed Particle Hydrodynamics (SPH) Meshing Techniques. Materwiss. Werksttech. 2020, 51, 550–557. https://doi.org/10.1002/mawe.201900222."
To provide a guide for readers, the following remark has been added to the introduction section of the revised manuscript (Page 2, footnote):
"*A brief background on the FEM simulation of the FSW thermal cycle is provided in Section 3.1."
Reviewer 2 Report
The manuscript presents the interfacial microstructure of AA6013/Ti-6Al-4V FSW welded parts and discusses the issues involving joining dissimilar materials. The following should be included to make the manuscript better before publication:
1. The heat conduction between the two surfaces is highly dependent on the interfacial condition. Were there any constraints (roughness, pressure, etc.) applied between the two materials regarding the condition of the interface?
2. If not, did the authors adjust any parameters to reproduce the validation results presented in Fig 2? What interfacial conditions did the experimental results put on (surface roughness, clamping pressure, etc.)?
3. In section 5.2, it is not clear whether the authors are stating that the formation of intermetallic compounds does not have any effect on cracking because no cracks were visible during the investigation. Is this correct? To be more convincing, I would recommend presenting the results by welding conditions.
The tense throughout the manuscript can be improved.
Author Response
Responses to the comments of Reviewer #2
on the paper entitled “Interfacial microstructure produced during dissimilar AA6013/Ti-6Al-4V friction-stir lap welding under zero-penetration condition” (Manuscript ID: metals-2614077)
The authors would like to express their gratitude to Reviewer for his/her remarks. Below, we provided specific replies to the issues raised.
Note: Reviewer’s comments are highlighted with bold.
The manuscript presents the interfacial microstructure of AA6013/Ti-6Al-4V FSW welded parts and discusses the issues involving joining of dissimilar materials. The following should be included to make the manuscript better before publication:
- The heat conduction between the two surfaces is highly dependent on the interfacial condition. Were there any constraints (roughness, pressure, etc.) applied between the two materials regarding the condition of the interface?
Authors’ response
In this study, no special precautions have been taken in order to precisely control the interfacial conditions. FSW was conducted in a typical (regular) manner. Accordingly, the welded blanks were ground with a water-abrasive paper and degreased with acetone, but the surface roughness was not measured. The evolution of Z-force was continuously recorded during FSW, but it was not controlled precisely. Thus, to address the Reviewer’s comment, the following remark has been added to the revised manuscript (Page 2, Section 2, Paragraph 2):
"The 2-mm-thick aluminum and titanium sheets were friction-stir lap welded using the AccuStir FSW machine. To provide a suitable quality for the surfaces to be welded, those were ground using sandpaper of 500 emery grit and then degreased with acetone... The magnitude of the Z-force during the steady welding stage was measured to be ~4.5 kN."
- If not, did the authors adjust any parameters to reproduce the validation results presented in Fig 2? What interfacial conditions did the experimental results put on (surface roughness, clamping pressure, etc.)?
Authors’ response
The fit between the simulation model and experimental measurements in Fig. 2 has been provided by adjusting the heat-transfer coefficient. To clarify this issue, the following remark has been added to the revised manuscript (Page 5, Paragraph 2):
"Given the sensitivity of the heat transfer process to the local variation of contact conditions (surface roughness, pressure, etc.), an elucidation of the heat-transfer coefficient is usually challenging. To avoid this difficulty, the coefficients were selected in order to provide the best fit between the experiment and the model. Specifically, it was found that the peak welding temperature was mainly determined by the heat-transfer coefficient between aluminum and titanium. On the other hand, the cooling rate was mainly influenced by the heat-transfer coefficient between titanium and the backing plate. In this way, the heat-transfer coefficient between the aluminum and titanium workpieces was taken to be 2,000 W(m2×oC), while that between the titanium and the backing plate was assumed to be 1,000 W(m2×oC)."
- In section 5.2, it is not clear whether the authors are stating that the formation of intermetallic compounds does not have any effect on cracking because no cracks were visible during the investigation. Is this correct? To be more convincing, I would recommend presenting the results by welding conditions.
Authors’ response
In order to avoid confusion, Section 5.2 has been removed from the revis
Reviewer 3 Report
The large block citation such as “[13,16-18,22,28,29].”, “[5,6,11,12,17,18,26,30,35-37]”; Should be avoided. Max 3 citations together are enough. Other citations like that in this manuscript should be amended too
It is not clear why the AA6013/Ti-6Al-4V combination were selected; please clearly indicate why
Please indicate how you managed to keep “was kept at ≈50 mm”
The sample preparation for ebsd should be described in details and also the measurement method too and post processing .
“Abaqus software.”- which version ?
“To save computational time, non-uniform” however from the Figure 1 it seems that the mesh is structured and therefore is uniform
The parameters listed in table 1 and 2 were determined by authors experimentally or taken from literature ? if the later one it requires citations
Figure 7 b is very low quality and difficult to understand the intermetallic and diffusion
Also the FEM do not shows nothing about intermetallic zone;
Even you claim that through temperature was higher at interface that temperature is not a melting temperature for aluminium therefore not very clear your claims. “..intermetallic compound” can be formed at higher temp ~600C and not 500 degrees
Not very clear also the claim about “Cracking resistance” as no pertinent details were provided
“Second, the fine-scale and discontinuous character..” were it is this as in your image not clear about this discontinuous as the image is low quality
na
Author Response
Responses to the comments of Reviewer #3
on the paper entitled “Interfacial microstructure produced during dissimilar AA6013/Ti-6Al-4V friction-stir lap welding under zero-penetration condition” (Manuscript ID: metals-2614077)
The authors would like to express their gratitude to Reviewer for his/her remarks. Below, we provided specific replies to the issues raised.
Note: Reviewer’s comments are highlighted with bold.
Note 2: To see the illustrations to the authors' responce, please check the attached Word file.
(1) The large block citation such as “[13,16-18,22,28,29].”, “[5,6,11,12,17,18,26,30,35-37]”; Should be avoided. Max 3 citations together are enough. Other citations like that in this manuscript should be amended too.
Authors’ response:
The authors do agree with Reviewer that three cited references are enough to confirm/support a statement. However, we suggest looking at this situation from another angle. During this study, the authors have checked all the scientific literature on the dissimilar FSW of aluminum and titanium alloys, which was available in the Scopus database. Thus, the reference list in this manuscript represents the entire collection of scientific literature on the subject. We do hope it might be useful for readers.
(2) It is not clear why the AA6013/Ti-6Al-4V combination were selected; please clearly indicate why
Authors’ response:
According to the comment, the following remark has been added to the revised manuscript (Page 2, Section 2, Paragraph 1):
"The program material employed in the preset study included commercial 6013 aluminum alloy and Ti-6Al-4V titanium alloy. These two materials are widely used in the transportation industry, and thus their practical usage should perhaps benefit from the application of FSW for their mutual joining. Moreover, the FSW behavior of these two alloys has been studied relatively well."
(3) Please indicate how you managed to keep “was kept at ≈50 mm”
Authors’ response:
According to the comment, the following remark has been added to the revised manuscript (Page 2, Section 2, Paragraph 3):
"…Furthermore, FSW was conducted under a plunge-depth control mode, and the distance between the probe tip and the titanium part was kept at ≈50 mm."
(4) The sample preparation for ebsd should be described in details and also the measurement method too and post processing.
Authors’ response:
According to the comment, the details of the EBSD procedure have been added to the revised manuscript (Page 3, Paragraphs 3 to 4):
"Microstructural examinations were conducted using scanning electron microscopy (SEM), energy-dispersive x-ray spectroscopy (EDS), and electron backscatter diffraction (EBSD). To obtain a suitable surface finish, microstructural samples were cut perpendicular to the welding direction of the welded joints and hot-mounted into an electro-conductive resin. Then, those were mechanically polished in conventional fashion using a series of water-abrasive papers and diamond pastes. The final polishing step comprised 24-hour vibratory polishing using commercial OPS suspension.
All microstructural observations were performed with an FEI Quanta 600 field-emission-gun SEM. The microscope was equipped with the TSL OIMTM system and operated at an accelerated voltage of 20 kV. EBSD maps were collected using a hexagonal scanning grid and a scan step size of 0.1 mm. For each diffraction pattern, nine Kikuchi bands were used for indexing to minimize errors. To avoid any modification of the experimental EBSD data, no post-processing (or clean-up) procedure was applied."
(5) “Abaqus software.”- which version ?
Authors’ response:
According to the comment, the software version has been indicated in the revised manuscript (Page 3, Section 3.1, Paragraph 1):
"…The thermo-mechanical 3-D model of dissimilar friction-stir lap welding of AA6013 and Ti-6Al-4V was elaborated using the commercial Abaqus 2020 software."
(6) “To save computational time, non-uniform” however from the Figure 1 it seems that the mesh is structured and therefore is uniform
Authors’ response:
In order to avoid misunderstanding, the text of the revised manuscript has been rephrased as follows (Pages 3 to 4):
"…To save computational time, a different mesh size was used within the stir zone and the remaining part of the aluminum workpiece. Specifically, the mesh size within the stir zone was 1.0×0.5×0.5 mm3 (length × width × thickness), while that in the remaining aluminum workpiece was 1.0×1.0×1.0 mm3 (Fig. 1)."
(7) The parameters listed in table 1 and 2 were determined by authors experimentally or taken from literature? If the later one it requires citations.
Authors’ response:
To clarify this issue, the text of the revised manuscript has been rephrased as follows (Pages 5 to 6):
"The input parameters for the Johson-Cook equation are summarized in Table 1. Except for the temperature exponent, the parameters were taken from the recent work by Saloomi [47]. The preliminary analysis showed that a value of = 1.34 (i.e., similar to that in Ref. [47]) leads to a significant overestimation of the Z-force. On the other hand, a relatively good fit between the calculations and measurements was obtained at = 0.4 (Table 1). It is suggested that this effect was associated with a replication of the real design of the welding tool (with a threaded tool probe) in the present study.
The physical properties of the model materials were taken from Refs. [57-60] and shown in Table 2."
References:
"[47] Salloomi, K.N. Fully Coupled Thermomechanical Simulation of Friction Stir Welding of Aluminum 6061-T6 Alloy T-joint. J. Manuf. Process. 2019, 45, 746–754. https://doi.org/10.1016/j.jmapro.2019.06.030.
[57] Boyer, R.; Welsch, G.; Collings, E.W. Material Properties Handbook: Titanium Alloys. eds. ASM International, Materials Park, OH, 1994
[58] Lu, X.; Lin, X.; Chiumenti, M.; Cervera, M.; Li, J.J.; Ma, L.; Wei, L.; Hu, Y.; Huang, W. Finite Element Analysis and Experimental Validation of the Thermomechanical Behavior in Laser Solid Forming of Ti-6Al-4V. Additive Manuf. 2018, 21, 30-40. https://doi.org/10.1016/j.addma.2018.02.003.
[59] Mechmet Ali, G.; Hasan, B.; Mustafa, C.; Selcuk, M. The Prediction of Surface Roughness and Tool Vibration by Using Metaheuristic-Based ANFIS during Dry Turning of Al Alloy (AA6013). J. Brazil. Soc. Mech. Sci. Eng. 2022, 44, 474. https://doi.org/10.1007/s40430-022-03798-z.
[60] Li, L.Q.; Hongbo, X.T.; Caiwang, M.N. Influence of Laser Power on Interfacial Microstructure and Mechanical Properties of Laer Welded-Brazed Al/Steel Dissimilar Butted Joint. J. Manuf. Proc. 2018, 32, 160-174. https://doi.org.10.1016/j.jmapro.2018.02002."
(8) Figure 7b is very low quality and difficult to understand the intermetallic and diffusion
Authors’ response:
Indeed, EBSD was not feasible to characterize the intermetallic layer. In the revised manuscript, this issue has been emphasized as follows (Page 9, Section 4.3, Paragraph 2):
"…Unfortunately, no indexable Kikuchi-patterns were detected within the intermetallic layer. Accordingly, this layer appeared black in the EBSD map, and no information on its internal structure was obtained (Fig. 7b)…"
Accordingly, the intermetallic layer was studied using scanning electron microscopy (Page 7, Section 4.1):
"A typical SEM micrograph taken from the welded surface between 6013 aluminum alloy and Ti-6Al-4V titanium alloy is shown in Fig. 4a. The development of a relatively thin (~0.5 mm), gray-colored layer at the joint interface is seen (highlighted by arrows). The magnified image of the interfacial layer is given in Fig. 4b.
|
|
|
Figure 4. (a) Typical SEM images of the welded surface between 6013 aluminum alloy and Ti-6Al-4V with selected area given at higher magnification in (b). Arrows exemplify interphase layer. ND, WD, and TD are normal direction, welding direction, and transverse direction, respectively |
In terms of the SEM contrast, this layer was broadly similar to the TiAl3 compound, which develops in Al/Ti sandwich structures during long-term annealing at 500 oC [e.g. 61]. In the present study, a notable characteristic of the interfacial layer was its complex topography (Fig. 4b). Of particular importance was the observation that the interfacial layer was discontinuous in nature (Fig. 4a)."
(9) Also the FEM do not shows nothing about intermetallic zone
Authors’ response:
Indeed, the FEM approach in the present study was applied only for the examination of temperature distribution within the weld zone but not for the development of intermetallic compound per se. This was due to the complexity of the underlying microstructural processes, which are still challenging for numerical simulations.
To clarify this issue, the following remark has been added to the revised manuscript (Page 3, Section 3, Paragraph 1):
"Considering the complexity of the underlying microstructural processes, the numerical simulation of the formation of intermetallic compounds during FSW is challenging. Hence, the FEM approach in the present study was applied only for a simulation of the thermal field generated within the weld zone during FSW. The obtained results were summarized in the present section."
(10) Even you claim that through temperature was higher at interface that temperature is not a melting temperature for aluminium therefore not very clear your claims. “..intermetallic compound” can be formed at higher temp ~600C and not 500 degrees.
Authors’ response:
In the scientific literature, there are several works reporting the formation of TiAl3 intermetallic compound at temperatures as low as 500 oC. For instance, please see “Liu, M.; Zhang, C.; Zhao, H.; Meng, Z.; Chen, L.; Zhao, G. Micro-/Nano-Scale Structure and Elemental Diffusion in the Al/Ti/Al Sandwich Structure. J. Mater. Res. Technol. 2023, 24, 9537-9552. https://doi.org/10.1016/j.jmrt.2023.05.204.”
To emphasize this issue, the text of the manuscript has been revised as follows (Page 7, Section 4.1, Paragraph 2):
"…In terms of the SEM contrast, this layer was broadly similar to the TiAl3 compound, which develops in Al/Ti sandwich structures during long-term annealing at 500 oC [e.g. 61]."
During dissimilar FSW of aluminum and titanium alloys, this intermetallic reaction has been reported to occur in a number of works. Please see the appropriate review “Simar, A.; Avettand-Fenoel, M.-N. State of the Art about Dissimilar Metal Friction Stir Welding. Sci. Technol. Weld. Join. 2017, 22, 389-403. doi: 10.1080/13621718.2016.1251712.”
(11) Not very clear also the claim about “Cracking resistance” as no pertinent details were provided
Authors’ response:
In order to avoid confusion, the section discussing this topic (i.e., Section 5.2) has been removed from the revised manuscript.
“Second, the fine-scale and discontinuous character..” were it is this as in your image not clear about this discontinuous as the image is low quality
Authors’ response:
Indeed, EBSD was not effective in characterizing the intermetallic compound, unfortunately. Thus, it was studied using conventional scanning electron microscopy. The typical results have been shown in Fig. 4 and summarized in Section 4.1 (Page 7 of the revised manuscript). The authors do hope that the discontinuous character of the intermetallic layer is seen in Fig. 4a.
"4.1. SEM observations
A typical SEM micrograph taken from the welded surface between 6013 aluminum alloy and Ti-6Al-4V titanium alloy is shown in Fig. 4a. The development of a relatively thin (~0.5 mm), gray-colored layer at the joint interface is seen (highlighted by arrows). The magnified image of the interfacial layer is given in Fig. 4b.
|
|
|
Figure 4. (a) Typical SEM images of the welded surface between 6013 aluminum alloy and Ti-6Al-4V with selected area given at higher magnification in (b). Arrows exemplify interphase layer. ND, WD, and TD are normal direction, welding direction, and transverse direction, respectively |
In terms of the SEM contrast, this layer was broadly similar to the TiAl3 compound, which develops in Al/Ti sandwich structures during long-term annealing at 500 oC [e.g. 61]. In the present study, a notable characteristic of the interfacial layer was its complex topography (Fig. 4b). Of particular importance was the observation that the interfacial layer was discontinuous in nature (Fig. 4a)."

Round 2
